# A Scaffold-Free 3-D Co-Culture Mimics the Major Features of the Reverse Warburg Effect In Vitro

**DOI:** 10.3390/cells9081900

**Published:** 2020-08-13

**Authors:** Florian Keller, Roman Bruch, Richard Schneider, Julia Meier-Hubberten, Mathias Hafner, Rüdiger Rudolf

**Affiliations:** 1Institute of Molecular and Cell Biology, Mannheim University of Applied Sciences, 68163 Mannheim, Germany; f.keller@hs-mannheim.de (F.K.); r.bruch@hs-mannheim.de (R.B.); m.hafner@hs-mannheim.de (M.H.); 2Institute of Medical Technology, Medical Faculty Mannheim of Heidelberg University and Mannheim University of Applied Sciences, 68167 Mannheim, Germany; 3TIP Oncology, Merck Healthcare KGaA, 64289 Darmstadt, Germany; richard.schneider@merckgroup.com (R.S.); julia.meier-hubberten@merckgroup.com (J.M.-H.)

**Keywords:** reverse Warburg effect, fibroblasts, MCT4, LC3, mitochondria, optical tissue clearing

## Abstract

Most tumors consume large amounts of glucose. Concepts to explain the mechanisms that mediate the achievement of this metabolic need have proposed a switch of the tumor mass to aerobic glycolysis. Depending on whether primarily tumor or stroma cells undergo such a commutation, the terms ‘Warburg effect’ or ‘reverse Warburg effect’ were coined to describe the underlying biological phenomena. However, current in vitro systems relying on 2-D culture, single cell-type spheroids, or basal-membrane extract (BME/Matrigel)-containing 3-D structures do not thoroughly reflect these processes. Here, we aimed to establish a BME/Matrigel-free 3-D microarray cancer model to recapitulate the metabolic interplay between cancer and stromal cells that allows mechanistic analyses and drug testing. Human HT-29 colon cancer and CCD-1137Sk fibroblast cells were used in mono- and co-cultures as 2-D monolayers, spheroids, and in a cell-chip format. Metabolic patterns were studied with immunofluorescence and confocal microscopy. In chip-based co-cultures, HT-29 cells showed facilitated 3-D growth and increased levels of hexokinase-2, TP53-induced glycolysis and apoptosis regulator (TIGAR), lactate dehydrogenase, and: translocase of outer mitochondrial membrane 20 (TOMM20), when compared with HT-29 mono-cultures. Fibroblasts co-cultured with HT-29 cells expressed higher levels of mono-carboxylate transporter 4, hexokinase-2, microtubule-associated proteins 1A/1B light chain 3, and ubiquitin-binding protein p62 than in fibroblast mono-cultures, in both 2-D cultures and chips. Tetramethylrhodamin-methylester (TMRM) live-cell imaging of chip co-cultures revealed a higher mitochondrial potential in cancer cells than in fibroblasts. The findings demonstrate a crosstalk between cancer cells and fibroblasts that affects cellular growth and metabolism. Chip-based 3-D co-cultures of cancer cells and fibroblasts mimicked features of the reverse Warburg effect.

## 1. Introduction

The human colon adenocarcinoma cell line HT-29 is frequently used as a model for colorectal cancer. Colorectal cancer is one of the most frequent malignancies in the Western world. Although annual rates declined over the last decades, in 2017, the life-time risk of disease in the USA was at 4.6% and 4.2% for men and women, respectively, with a death rate of roughly 38% [1]. Upon surgical removal of colorectal adenoma, 3% of patients develop colorectal cancer within a median follow-up of 7.7 years [2]. Thus, effective treatment options are still needed. To decrease the median development time and cost, efficient pre-clinical screening methods are being investigated by researchers and developers worldwide [3].

Like in many other malignancies, colorectal cancer also exhibits an extraordinarily high demand of energy, which is due to excessive metabolic activity and proliferation [4]. Therefore, total energy intake and macronutrient consumption serve as risk indicators [5]. Furthermore, the characteristic exuberant glucose uptake of colorectal lesions has been regularly exploited for diagnostic purposes in clinical molecular imaging by using ^18^F-fluorodeoxyglucose as a biomarker to identify metastatic sites [6]. However, metabolic interactions within tumors are complex and involve heterogeneous tumor cell populations ranging from anabolic to catabolic forms that must be considered for the establishment of reliable cancer models [7]. Furthermore, stromal components, such as mesenchymal cells, might also participate in the metabolic interplay [8]. While the increased overall glucose consumption in a tumor indicates an altered metabolic activity with a tendency towards aerobic glycolysis [9], it is debated if all cells in a tumor undergo such a metabolic switch, or whether this occurs primarily for either tumor or stromal cells. According to the Warburg hypothesis, it is the cancer cells that do so [10]. However, this is in conflict with the observation that in situ, tumor cells often display intact mitochondria [11]. Thus, a reverse Warburg hypothesis was formulated that proposes the occurrence of a signaling from the cancer cell to the stromal fibroblast, which modulates the metabolic behavior in the fibroblast and induces its shift to aerobic glycolysis [12]. This may lead to the production of reactive oxygen species and a general change in redox status. Accordingly, extensive alterations within tumor tissue sites [13], such as angiogenesis, migration, acidification, and hypoxia, are the consequence [14].

While cancer cells eventually enter a neoplastic state through reduced restrictions of proliferative potential [15], a metabolic shift of fibroblasts can bring these cells from a suppressive to a cancer-supportive state [16]. These stromal cells are usually referred to as cancer-associated fibroblasts (CAFs) and exhibit a strongly altered expression profile [17], including overexpression of smooth muscle actin or vimentin [18] and downregulation of caveolin 1. In addition, CAFs show enhanced autophagic activity, leading to higher expression levels of microtubule-associated proteins 1A/1B light chain 3 (MAP1LC3, short LC3) and ubiquitin-binding protein p62 (P62), both markers for autophagosome formation [19], and a loss of mitochondria through mitophagy [20]. This further favors a shift to aerobic glycolysis and the accumulation of short carbon-rich metabolites like lactate, which can be shuttled to surrounding cancer cells [21]. Therefore, mono-carboxylate transporters (MCT) play an important role [22], with MCT4 and MCT1 being principal exporters and importers, respectively. While MCT4 is often upregulated in CAFs, MCT1 expression is typically abundant in tumor cells [23]. The increased lactate levels can enhance metabolic activity in oxidative cancer cells [24,25], and the metabolic plasticity of tumor cell mitochondria is essential for adequate processing of nutrients available on a non-regular basis [26]. Such metabolic interactions can be instrumental for tumorigenesis, drug resistance, and tumor relapse [24].

Current metabolic models often either originate from primary tissue biopsies [27] or include xenograft tissues [28] relying on Matrigel or basal membrane extract (BME) as the supporting material [29]. However, a major issue here is that approaches based on primary cells are time consuming, expensive, and prone to low reproducibility due to the intrinsic variability of the patient material itself. On the other hand, commonly used two-dimensional mono-cultures often lack important components of intercellular signaling, which are crucial for drug testing [30]. Therefore, Matrigel, as a naturally derived hydrogel scaffold for 3-D cell cultures that are more homogeneous compared to biopsies or xenografts, has been the gold standard for a wide range of cell types [31]. These hydrogels provide biophysical cell-adhesive properties and can contribute to cell viability and growth [32], but they also contain an ill-defined mixture of growth factors that influences cellular behavior in an unpredictable way due to batch-to-batch variances [33], potentially obliterating the examination of delicate alterations in short-term cultures.

We aimed to model the reverse Warburg effect using 2-D and 3-D co-cultures of human HT-29 colon cancer cells and human CCD-1137Sk fibroblasts. While Dynarray cell chip microarray cultures reflected major aspects of the reverse Warburg effect, 2-D cultures partially reproduced these findings but lacked effects on HT-29 cells. Conversely, spheroid co-cultures did not show enhanced growth, suggesting a lack of an efficient metabolic advantage under this condition.

## 2. Materials and Methods

### 2.1. Cell Culture

CCD-1137Sk human fibroblasts were kept in Iscove’s Modified Dulbecco’s Medium (Capricorn; IMDM-A; Lot #CP18-2245) supplemented with 10% fetal bovine serum (Capricorn; FBS-12B; Lot #CP16-1422), and 1% penicillin/streptomycin (Capricorn; PS-B; Lot #CP18-2207). HT-29 human colon cancer cells were cultivated in McCoy’s 5a media (Capricorn; MCC-A; Lot #CP19-2689), supplemented with 10% FBS-12B and 1% PS-B. The media for MDA-MB-231 human breast cancer cells consisted of Dulbecco’s Modified Eagle Medium (Capricorn; DMEM-HPA; Lot #CP18-2096) supplemented with 10% FBS-12B, 1% PS-B, and 1% Minimum Essential Medium Nonessential Amino Acids (Capricorn; NEAA-B; Lot #CP17-1726). All cell lines were passaged twice per week with a seeding density of 1 × 10^6^ cells/T75 flask. Two-dimensional cell culture analysis was based on cover slips (12 mm; VWR; ECN 631-1577; Lot #43395 819) placed in cell culture dishes (Greiner; PS; 664 160) before seeding a total of 1.2 × 10^6^ cells in appropriate ratios and culturing for four days.

### 2.2. Spheroid Formation

To analyze growth behavior, the spheroid formation was performed with a total of 6000 cells seeded per well within different mono- and co-culture ratios using 96-well cell-repellent microplates (Greiner; 650970; Lot #E2001347). The BME dependency of the spheroid metabolism was analyzed by setting 250 HT-29 or 4000 CCD-1137Sk cells in a mono-culture condition. Additionally, 250 cancer cells were co-cultivated together with 500 fibroblasts. This ensured a more homogeneous spheroid mean diameter distribution around 300 µm after four days, preventing size-dependent effects from influencing the cellular biology approach. Appropriate amounts for 2.5% or 10% BME dissolved in media were added immediately after the culture suspensions. Cell aggregation was enhanced by centrifugation for 6 min at 500 rcf.

### 2.3. Dynarray Preparation

For Dynarray (300MICRONS; Dynarrays MCA-C300-300-PC) seeding, the microarrays were sterilized with 100%, 70%, 50%, and 30% isopropanol for 1 min, respectively, before washing twice in sterile deionized water. Then, 100 µg/mL Type-I collagen (Roche; rat tail Collagen; 11179179001) dissolved in 150 µL of sterile deionized water was incubated for 1 h on top of the cavities at room temperature for coating the microarray. The cancer cells and fibroblast cells were seeded at appropriate ratios to result in a density of 4000 cells per cavity. The Dynarrays were placed in cell culture dishes and after 4 h at 37 °C and 5% CO_2_, 10 mL of media were added. After fixation in 4% PFA/PBS, Dynarrays were either stained, cleared, and imaged as whole mounts or orthogonal Dynarray slices were made. In brief, chips were embedded in 2% agarose (Roche; Agarose MP; 11 388 983 001) dissolved in Tris-acetate-EDTA (TAE) buffer (0.04 M tris base, 0.02 M acetic acid, 0.01 M EDTA) and 50-µm-thick slices were collected using a vibratome (Leica Biosystems; Vibratome Model VT1000S). To label the cells with a CellTracker, mono-type suspensions of either HT-29 or CCD-1137Sk cells were incubated for 1 h in fetal bovine serum (FBS)-free media (37 °C/5% CO_2_) and washed with PBS directly before co-seeding. For live-cell analysis, Dynarray cultures were incubated in Tetramethylrhodamin-methylester (TMRM)-containing complete media (30 min/37 °C), washed 3 times with PBS, and transferred into 25 mM -(2-hydroxyethyl)-1-piperazineethanesulfonic acid (HEPES) buffer to ensure steady pH conditions throughout the imaging process.

### 2.4. Antibodies and Dyes

Primary antibody incubation was performed in a static incubation (ON/4 °C) for 2-D and on a roller mixer (ON/37 °C) for 3-D samples. Secondary antibodies were used for 1 h at room temperature for flat and as the primary antibodies for spheroid and Dynarray preparation. Antibody or dye details and concentrations were as follows: Anti-Carcino Embyonic Antigen CD66e (CEA; Thermofisher scientific; MIC0101; Lot #VD2969991; dilution 1:250), Anti-Type-I Collagen (Collagen 1; Rockland; 600-401-103-0.5; Lot #41250; dilution 1:100), Anti-Type-IV Collagen (Coll4; Rockland; 600-401-106-0.5; Lot #40995; dilution 1:100), CellTracker Deep Red (CTdr; Lifetechnologies; C34565; Lot #1781143; stock 2 mM; dilution 1:1000), CellTracker Green (CTg; Lifetechnologies; C7025; Lot #1913919; stock 10 mM; dilution 1:1000), 4’,6-Diamidino-2-Phenylindole (DAPI; Roche; 10236276001; Lot #28114320; 1 mg/mL; dilution 1:1000), Anti-Hexokinase II (HK-2; abcam; ab209847; Lot #GR3220265-2; dilution 1:100), Anti- Microtubule-associated proteins 1A/1B Light Chain 3 (LC3; CellSignalling; 3868S; Lot #11; dilution 1:200), Anti-Lactate dehydrogenase (LDH; abcam; ab47010; Lot #GR3307117-1; dilution 1:1000), Anti-Monocarboxylate Transporter 1 (MCT1; Sigma-Aldrich; Anti-SLC16A1; Lot #C114705; dilution 1:500), Anti-Monocarboxylate Transporter 4 (MCT4; SantaCruz Biotechnology; sc-376140; Lot #D3018; dilution 1:400), Anti-Ubiquitin-binding protein p62 protein (P62; Progen; GP62-C; Lot #703241-03; dilution 1:100), Anti-Succinate Dehydrogenase (SDH; Thermofisher scientific; 459200; VC296863; dilution 1:200), Anti-TP53-induced glycolysis and apoptosis regulator (TIGAR; abcam; ab37910; Lot #GR3210648-2; dilution 1:500); Anti-Tetramethylrhodamine (TMRM; Invitrogen; T-668; 500 µM; dilution 1:2000), Anti-Translocase of outer mitochondrial membrane 20 (TOMM20; Sigma-Aldrich; HPA011562-100UL; Lot #4450; dilution 1:400), Goat α-mouse AF 488 (M; Invitrogen; A11001; Lot #1834337; dilution 1:1000), Donkey α-rabbit AF 488 (R; Invitrogen; A21206; Lot #176375; dilution 1:1000), Goat α-guinea pig AF 555 (GP; Invitrogen; A21435; Lot #2015338; dilution 1:1000), and Goat α-rabbit AF 647 (R; Invitrogen; A21246; Lot #55002A; dilution 1:1000).

### 2.5. Data Acquisition

Brightfield (Zeiss Axiovert 25/objective CP-ACHROMAT, 5x/0.12Ph0) and confocal (Leica SP8, objectives HC PL FLUOTAR 10 × /0.30, HC PL APO 20 × /0.75 IMM CORR and HC PL APO CS2 63 × /1.2 W CORR) microscopy and the evaluation of spheroid size [34] were conducted as published earlier [35]. For confocal analysis, pictures with a resolution of 1024 × 1024 pixels w×ere taken with 1-µm z-steps for 3-D stacks. Whole-mount staining was enhanced by a clearing procedure based on the glycerol clearing protocol as published [36]. Briefly, samples were immersed in an aqueous glycerol solution with a refractive index of 1.459 containing 88% of glycerol (ON/37 °C) on a roller mixer. Microscopy was performed in the same solution. Therefore, samples were fixed in 4% PFA/PBS, treated with 2% Triton X-100 (2 min/RT), incubated in penetration buffer (0.2% Triton X-100, 0.3 M glycine, 20% DMSO in PBS/30 min/37 °C), blocked with bovine albumin serum (1% BSA, 0.2% Triton X-100, 10% DMSO in PBS/2 h/37 °C), and stained with antibodies incubated in appropriate buffer (0.2% Tween 20, 10 µg/mL heparin, 1% BSA, 5% DMSO in PBS/ON/37 °C). For intermediate washing steps, a buffer containing 0.2% Tween 20 and 10 µg/mL heparin in PBS was used.

### 2.6. Data Analysis

Two-dimensional image analysis and assembly were done with ImageJ software (v1.48v) and based on signal intensities in split channels. In 2-D, regions of interest were set based on: 4′,6-diamidino-2-phenylindol (DAPI) and the carcinoembryonic antigen (CEA) channel threshold.

For 3-D image analysis and corresponding data visualization, a dedicated Python script was established. Each stack was divided into a set of small non-overlapping regions with x, y, and z dimensions of 32, 32, and 16 voxels, respectively. To minimize the influence of background signals, the images were first segmented based on their nuclei channel. Therefore, the DAPI signal of an image volume was smoothed using three-dimensional Gaussian blur and then segmented using a threshold value calculated by Otsu’s method [37]. To also include adjacent regions outside the nuclei, high sigma values of 4, 4, and 0 for the x, y, and z dimension were used for Gaussian smoothing. An exemplary sum-projection of a segmentation including the visualization of the regions’ borders can be seen in Figure 6C. Only regions with more than 50% foreground based on segmentation were included. For the remaining regions, the average foreground intensity of the voxels was calculated for DAPI, CEA, and the corresponding marker channel. For visualization, composite results of 5 cavities for each culture type were assembled.

Statistics were calculated with GraphPad Prism 7. For direct comparison of two data points within the immunofluorescence evaluation, significance was defined based on the *p*-values (* *p* ≤ 0.05; ** *p* ≤ 0.01; *** *p* ≤ 0.001; **** *p* ≤ 0.0001) in a multiple comparison t-test analysis without assumptions. Normal distribution and homoscedasticity were tested with the Kolmogorov–Smirnov procedure. For testing of co-localization, the Coloc2 algorithm of ImageJ was used with a PSF of 3.0 and 10 randomizations. The Pearson’s correlation (R) above threshold was evaluated. Furthermore, one-way ANOVA with Holm–Sidak multiple comparison was performed for the growth curve comparisons. Significance was defined based on *p*-values (* *p* ≤ 0.05; ** *p* ≤ 0.01; *** *p* ≤ 0.001; **** *p* ≤ 0.0001).

## 3. Results

### 3.1. MCT4 as well as Markers for Glycolysis and Autophagy are Upregulated in CCD-1337Sk Fibroblasts upon Co-Culture with HT-29 Cells

First, we investigated whether co-cultures of HT-29 cells and CCD-1137Sk fibroblasts exhibited an altered expression of lactate transporters as compared to the respective mono-cultures. Therefore, 2-D mono- and co-cultures were set up and cultured at a confluency of up to 80% for four days. Then, immunofluorescence staining was first done for MCT1 and MCT4, as markers for lactate influx and efflux, respectively. While both MCT1 and MCT4 were expressed more strongly in HT-29 tumor cells than in fibroblasts, only MCT4 increased significantly in the fibroblasts upon co-cultivation (Figure 1A–C). Conversely, MCT1 remained low in fibroblast cells under all conditions (Figure 1A–C). Discrimination between HT-29 and CCD-1137Sk cells was done on the basis of three criteria. First, HT-29 consistently grew in dense islets, both in mono- and co-culture, whereas CCD-1137Sk typically showed a spindle-shaped morphology and grew in between the HT-29 islets in the co-cultures. Second, the molecular markers, carcinogen embryonic antigen (CEA) [38] and collagen 4 (Coll4) [39], were primarily expressed in either HT-29 or CCD-1137Sk cells, respectively (Appendix A). These served as additional confirming features for the cell-type selection. Finally, DAPI staining of CCD-1137Sk cell nuclei was mostly darker than that of HT-29 cells and showed a more elongated and larger area. This trait was also used for the later analyses of the 3-D data sets.

Next, the effects of the co-culturing of HT-29 and CCD-1137Sk cells on their metabolic profiles were addressed. Therefore, immunofluorescence staining of the same cultures as those mentioned in Figure 1 was done for hexokinase-2 (HK-2), lactate dehydrogenase (LDH), TP53-induced glycolysis and apoptosis regulator (TIGAR), succinate dehydrogenase (SDH), and translocase of outer mitochondrial membrane 20 (TOMM20), as markers for glucose breakdown, pyruvate-lactate metabolism, negative glycolysis regulation, oxidative phosphorylation, and mitochondrial content, respectively (Figure 2A–C) [4]. While HT-29 cells did not show any significant change in any of these markers, CCD-1137Sk cells displayed altered expression levels consistent with an upregulation of glycolysis and a downregulation of oxidative phosphorylation. Indeed, HK-2 went up, whereas TIGAR and TOMM20 decreased. LDH and SDH remained unaltered under these conditions.

The data on lactate transporters and metabolic markers were consistent with a co-culture-induced switch of the CCD-1137Sk cells towards a catabolic phenotype. To further corroborate this possibility, the expression of markers for autophagy, namely LC3 and P62, were assessed under mono- and co-culture conditions. As shown in Figure 3, both proteins were upregulated in CCD-113Sk in co-culture with HT-29 cells. Conversely, no alteration of LC3 or P62 levels was found in HT-29 cells upon co-culturing with fibroblasts.

Finally, to investigate whether the observed modifications of fibroblast metabolism were exclusively related to the co-culture with HT-29 cells, analogous experiments were carried out with the human breast cancer cell line MDA-MB-231 in mono- and co-culture with CCD-1137Sk fibroblasts. As shown in Appendix A, the overall results were essentially the same: Upon co-culture with MDA-MB-231 cells, CCD-1137Sk cells showed enhanced levels of MCT4, HK-2, LC3, and P62 and reduced presence of TIGAR and TOMM20. In contrast, the breast cancer cells did not show any such alteration.

### 3.2. Addition of CCD-1137Sk Fibroblasts does not Alter Growth of HT-29 Spheroids

Although 2-D cultures are ideal for quantitative analyses, they often lack gradients of drugs, oxygen, waste products, and nutrients as they are present in vivo. Therefore, we next established three-dimensional spheroid co-cultures with CCD-1137Sk fibroblasts and HT-29 colon cancer cells. To test the best conditions for a homogeneous distribution of both cell types in spheroids, different ratios of cancer cells and fibroblasts were either seeded together or sequentially in ultra-low attachment plates and allowed to grow for up to 30 days. Every seeding procedure resulted in rather round spheroids. Since the condensation of the first cell population into a spheroid was typically accomplished within two days (Figure 4A), the secondary seeding in the sequential protocols was performed on that day. Size measurements of the mean diameters revealed that neither the different tested cell-line ratios, nor seeding sequences altered the assembly or growth of spheroids (Figure 4B). Indeed, all spheroids with cancer cells alone or cancer cells plus fibroblasts showed a very similar behavior: After an initial condensation phase, which led to a minimal mean diameter of spheroids at two to three days, they followed an asymptotic growth curve and reached a maximal spheroid diameter of approximately 1000 µm (Figure 4B). Thus, CCD-1137Sk fibroblasts did not contribute to any major change in the growth of HT-29 spheroids. Accordingly, fibroblast mono-culture spheroids remained small and did not increase in size over time (Figure 4B). An examination of the distribution of Coll4 and CEA proteins on optical slices from co-culture spheroids suggested that fibroblasts remained rather confined to small islets (Figure 4C, arrowheads). Thus, it appeared that there was limited cross-talk between both cell types, suggesting that the spheroid model was not ideal to study mutual interactions between HT-29 cancer cells and CCD-1137Sk fibroblasts.

### 3.3. Growth of CCD-1137Sk and HT-29 Co-Cultures is Enhanced in Dynarray Chips

Reasoning that the spherical arrangement with its observed strict spatial separation between HT-29 and CCD-1137Sk cells might have contributed to the lack of an effect of co-culturing on spheroid growth, we switched to chip-based 3-D Dynarrays. These have cavities of 300 µm in diameter and depth and their inner surfaces can be coated with extracellular-matrix components or gels. Furthermore, the walls of the cavities are porous, allowing the passage of nutrients. HT-29 and CCD-1137Sk cells were either seeded as mono- or co-cultures into collagen-1-coated Dynarray chips and cellular growth was assessed. In all cases, 4000 cells were seeded per cavity, i.e., either 4000 HT-29 or CCD-1137Sk cells in the mono-cultures or 2000 cells of each type for the co-cultures. In this format, the fibroblast marker collagen 1 was widely distributed throughout the co-cultures (Figure 5A), suggesting a better mixing of both cell types than in spheroids. MCT4 fluorescence signals were comparable in both HT-29 and CCD1137-Sk cell areas of the co-culture chips (Figure 5A). Furthermore, the growth behavior of HT-29 cells in Dynarray chips was influenced by the presence or absence of fibroblasts. While HT-29 cells adhered in the mono-culture to the inner chip walls and grew in only few flat layers (Figure 5B), co-cultures with fibroblasts filled out the entire chip cavities and even often grew beyond (Figure 5B). Total cell counts increased in co-cultures by 55% as compared to HT-29 mono-cultures after four days in culture (Figure 5C). This demonstrates that the growth behavior of HT-29 and/or CCD-1137Sk cells can be modulated by 3-D co-culturing and that this is dependent on the type of 3-D culture system.

### 3.4. Fluorescence Staining of CEA, Coll4, and DAPI Shows Different Intensities in HT-29 and CCD-1137Sk Cells

To assess the potential effects of co-culturing on metabolic markers in Dynarray cultures of HT-29 and CCD-1137Sk cells, we sought cell-type-specific features that could later serve as discriminators. Thus, in analogy to the 2-D cultures, chips were stained with DAPI, anti-CEA, and anti-Coll4, and were then visualized by 3-D confocal microscopy. This revealed a high density of nuclei (Figure 6A), impairing an instance segmentation of individual cells. Yet, a correlative quantitative voxel analysis could be implemented to determine the relative staining intensities. Therefore, 3-D image stacks were subdivided into voxel cubes, as shown in Figure 6C, and the average fluorescence intensity per cube was determined after nuclei segmentation and background subtraction. This showed the following principal features. First, in mono-cultures, HT-29 cells showed brighter CEA and darker Coll4 signals than CCD-1137Sk cells (Figure 6B). In this and all following graphs, a red vertical line marks the CEA brightness range that contains 95% of all CCD-1137Sk voxel cubes in mono-culture. Second, also in mono-cultures, DAPI fluorescence was much brighter in HT-29 cells than in CCD-1137Sk cells (Figure 6B, color code). Third, in co-cultures, the Coll4 vs. CEA distribution looked more similar to the HT-29 mono-culture than to the CCD-1137Sk mono-culture distribution (Figure 6B), suggesting that most cells in co-cultures were HT-29 cells. However, a limited number of signals in the lower left quadrant of the co-culture distribution showed a combination of low DAPI and CEA signals on the one hand and higher Coll4 signals on the other hand, suggesting that these could indicate the fibroblast population (Figure 6B). Altogether, a picture emerged that using this kind of voxel-based analysis, the two cell populations could be differentially assessed with respect to their marker expression, at least under the assumption that in co-cultures the marker expression did not alter in a confounding manner. However, it also appeared that either the fibroblast population in the co-cultures was rather small and/or that signals from HT-29 cells were preferentially detected in co-cultures using this approach. In the following, we used CEA and DAPI staining as discriminators to address the expression of metabolic markers in the two cell types.

### 3.5. Fluorescence Signals of Metabolic Marker Proteins Show Differential Regulation in Dynarray Co-Cultures

Given that the voxel-cube analysis was promising to distinguish relative marker-protein expression levels in 3-D cultures, we wanted to assess a potential metabolic cross-talk between HT-29 and CCD-1137Sk cells in chips. Therefore, Dynarrays were seeded with mono-cultures of HT-29 or CCD-1137Sk cells or with co-cultures of both cell lines as described before and grown for four days. Then, chips were fixed, cleared, and stained with DAPI and anti-CEA to segregate the cell types from each other. In addition, the samples were labeled with antibodies against metabolic marker proteins HK-2, LDH, TIGAR, or TOMM20. Three-dimensional confocal microscopy of stained specimens was done, and image stacks were quantitatively analyzed using the voxel-cube approach. As already evident from simple sum-z projections (Figure 7A), all metabolic-marker proteins exhibited different intensities, within a given culture, between the two cell types, and in the mono- vs. co-culture condition. The quantitative analysis of all individual voxel cubes gave the following major results: First, comparing the mono-cultures of HT-29 and CCD-1137Sk, CEA signals always extended more to the brighter value range for HT-29 than for CCD-1137Sk cells; in co-cultures, the CEA brightness was further right-shifted. Second, for the markers HK-2 and TIGAR, the fluorescence distribution in HT-29 mono-cultures was bimodal with a low- and a high-intensity cluster, suggesting the presence of two subpopulations of cells with either a low or high abundance of HK-2 and TIGAR; this did not correlate to the expression of CEA. In co-cultures, the same kind of bimodal distribution was found, but here, both CEA and the metabolic marker protein fluorescence signals were higher, leading to an overall shift of the data cloud to the upper-right part of the graphs; this was more pronounced for HK-2 than for TIGAR. Third, CCD-1137Sk cells in mono-cultures showed no bimodal distribution for any of the markers but rather displayed a continuous range of fluorescence intensities for all four tested markers. However, while TIGAR and LDH reached high values in CCD-1137Sk mono-cultures compared to HT-29 mono-cultures, the intensities of the HK-2 and TOMM20 signals were similar or lower in CCD-113Sk than in HT-29 cells. Finally, while LDH and TOMM20 were apparently going up in the HT-29 population of co-cultures (right from the red line), TIGAR and LDH were seemingly downregulated in CCD-1137Sk cells upon co-cultivation. Though, the latter effect might have also been caused by a potential skew towards HT-29 cells of the analysis method.

### 3.6. Autophagy is Enhanced in Dynarray Co-Cultures of HT-29 and CCD-1137Sk Cells

Next, the voxel-cube approach was used to assess the potential regulation of autophagy in 3-D chip cultures. Therefore, mono- and co-cultures of HT-29 and CCD-1137Sk cells in Dynarrays were fixed after four days, cleared, and stained with DAPI and anti-CEA as before. To study the presence of autophagy, samples were co-labeled with anti-LC3 or anti-P62. Although a first look on the sum-z projections suggested that the fluorescence intensity of both LC3 and P62 was higher in co-cultures than in mono-cultures of either HT-29 or CCD-1137Sk (Figure 8A), the voxel-cube analysis did not confirm this. Rather, it seemed that the brightness distributions of both LC3 (Figure 8B) and P62 (Figure 8C) were rather similar when comparing HT-29 mono-cultures and co-cultures. For HT-29 mono- and co-cultures, LC3 signals showed a bimodal distribution, similar to the ones observed for HK-2 and TIGAR. Conversely, CCD-1137Sk cells in mono-culture again showed simple continuous ranges of LC3 and P62 fluorescence intensities. Since the lack of higher LC3 and P62 values in co-cultures upon voxel-cube analysis did not reflect the sum-z projections well and there was suspicion that the voxel-cube approach might have a skew towards HT-29 signals, the data were additionally analyzed by fluorescence intensity measurements per µm^3^ regardless of cell type. As depicted in Figure 8D, this corroborated the qualitative impression of the sum-z projections. Indeed, using this approach, LC3 and P62 signals were highest in the co-cultures and lowest in the CCD-1137Sk mono-cultures. Of course, this did not allow a cell-type-specific statement, but in consideration of the whole picture, a major contribution of fibroblasts to the high autophagy marker values in co-cultures appeared likely.

### 3.7. In Microarray Chip Co-Cultures, CCD-1137Sk Fibroblasts Show Low Mitochondrial Membrane Potential

Cells undergoing Warburg- or reverse Warburg-based metabolic switches are expected to vary greatly with respect to mitochondrial activity. Thus, to test the presence of active mitochondria, live cell imaging was performed using the red-fluorescent vital dye TMRM. This accumulates in mitochondria depending on their membrane potential. To be able to differentiate between the cell types in live cell imaging, CCD-1137Sk and HT-29 cells were loaded with either CellTracker DeepRed (CTdr) or CellTracker Green (CTg), respectively. Then, they were co-cultured in Dynarray chips for five days. Next, 30 min after loading with TMRM, 3-D live cell confocal microscopy was performed. At this point, it is necessary to say that in the control experiments, the CellTracker staining itself also enhanced LC3 expression primarily in CCD-1137Sk cells of HT-29 co-cultures (data not shown). Therefore, we cannot completely exclude that the TMRM data shown here were partially affected by the experimental procedure itself. However, as observed upon immunostaining against collagen 1 and CEA (see Figure 5A), the CellTracker signals also revealed clusters of fibroblasts interspersed within the HT-29 colony (Figure 9A). While CTg-positive HT-29 cells exhibited variable but mostly bright TMRM fluorescence, CTdr-positive CCD-1137Sk cells were almost completely devoid of such signals. Co-localization analysis of TMRM with the labeling intensity for each CellTracker dye using Pearson’s coefficient measurements confirmed this observation (Figure 9B). This suggests that HT-29 cancer cells were present in different states of mitochondrial activity, while fibroblasts were mostly lacking a consistent mitochondrial membrane potential. Moreover, additional immunostaining of co-culture chips showed that regions outside the circular entry of the chip cavities, which were primarily populated by fibroblasts, were strongly positive for MCT4 but not for MCT1 (Figure 10). Conversely, the circular body within the 3-D cavities, which was composed of mixed HT-29 and CCD-1137Sk populations, exhibited variable amounts of MCT1 and MCT4, showing a prevalence for either one or the other kind of transporter.

## 4. Discussion

Since a few years ago, mutual interactions between cancer and stromal cells have emerged as major contributors to tumor metabolism [40]. Paracrine and autocrine signaling of growth factors, proteases, inhibitors, and extracellular matrix components might lead to widespread cross-talk between cancer cells and their surroundings. This influences the metabolism of all cells involved [41] and has become a widely studied target for cancer treatment. However, cell model systems with desired properties in terms of both fast and easy usability, as well as sufficient physiological complexity, including effects within tumor populations of cancer and stromal cells for drug treatment investigation, are rare.

Current in vitro model systems for cancer studies mostly employ either cell lines or primary tumor cells and these are either cultured on planar surfaces or, using different methods, in 3-D. Regarding the choice of ‘cell line’ versus ‘primary cells’, the latter appears superior for personalized medicine [42] and for a higher physiological relevance [43]. However, since primary cells are derived from a limited amount of tumor material and since the genotype and phenotype are variable not only between any two patients but also within the tumor population of a given patient, primary cells exhibit major inherent variability in terms of availability and genetic constitution [44]. Therefore, while in vitro models derived from primary cells might be advantageous for personalized medicine and advanced stages of drug testing, the use of cell lines can still be an option for the study of more fundamental cell biological phenomena and principles of cellular signaling due to their unlimited availability and relative phenotypic robustness. With respect to culturing in 2-D or 3-D, the former is still widely used due to its unbeatable applicability and analytical amenability, in particular, in the context of image-based assays [45]. Nevertheless, signaling processes mediated by cell–cell contact or paracrine mechanisms might be reliably addressed in such adherent cell cultures. However, since contact via the entire cell surface as well as the establishment of physiological cell organization, of gradients of drugs, oxygen, waste products etc. can only be obtained in 3-D cell cultures, the latter has increasingly gained importance in recent years. Therefore, spheroids and, particularly, tumor organoids have become key to in vitro cancer studies [46]. Spheroids are mostly based on the self-assembly of immortalized cells by impairing their settlement on any non-cellular surface. In contrast, tumor organoids involve the use of primary tumor-derived cells and their embedding in hydrogels formed by extracellular matrix extracts, typically Matrigel, BME, or similar. Models of intermediate complexity use Matrigel in combination with cell lines, in particular, for obtaining 3-D cultures from cell lines that otherwise hardly form such cultures [35,47]. Although Matrigel/BME-based tumor organoids have been seen as a ‘gold standard’ for in vitro cancer research, the use of Matrigel also has important drawbacks. For example, there is a considerable lot-to-lot variability, Matrigel is derived from tumor tissue, and it inherently adds specific physicochemical matrix properties [48]. Furthermore, it has a significant autofluorescence in the green wavelength range [49,50], rendering its use in marker-specific microscopic analyses less efficient. We compared 3-D cultures made with and without BME and experienced issues of autofluorescence. On top, BME-based cultures were much less regular and condensed, leading to an increased size of 3-D data stacks, an issue that requires more efforts on data acquisition time, storage, and analysis (Appendix A). Most worrying, though, were the differences observed in terms of marker expression. For example, in CCD-1137Sk mono-cultures, the presence of BME increased the fluorescence signals obtained for CEA and decreased those for LC3 (Appendix A).

Recapitulating, different model systems might be superior to others for certain aspects, but the choice of an appropriate condition will likely depend strongly on the research topic. In this context, here, we used yet another assay system, i.e., the Dynarray chip. From a purely technological point of view, this can be an interesting option for mechanistic studies where both intrinsic cell–cell interactions as well as a defined surface composition should be guaranteed. These chips feature equally spaced and equally sized cavities made by an optically transparent polymer, which can be coated with any material of interest, e.g., Matrigel, BME, collagen, or also just poly-L-lysine, to enhance interaction between the chip and cells. We chose collagen 1 to allow for a defined support and to avoid potentially interfering signaling molecules that might come with Matrigel/BME. Indeed, at least in 2-D cultures, coating with collagen 1 did not affect the expression of any of the tested markers at the level of immunofluorescence (Appendix A). In Dynarray co-cultures of HT-29 or MDA-MB-231 cells with CCD-1137Sk fibroblasts, the latter were typically making most of the contacts to the collagen surface and the morphology of these peripheral cells was often spindle shaped, forming bridges between the chip wall and the mass of the co-culture. Conversely, in the centers of the cavities, cells were better mixed, although some segregation of fibroblasts and tumor cells into nests was also observed, like in the spheroids. Altogether, this mediated a high homogeneity of 3-D cultures between different cavities and different chips. Another technical advantage of the chips over the spheroid cultures was their spatial regularity, which eased automated microscopic acquisition and subsequent data analysis.

Regarding the metabolic interactions, the 2-D cultures and Dynarrays revealed good consensus for fibroblasts. Indeed, in both systems, data were consistent with a higher level of HK-2 and autophagy markers as well as a loss of TIGAR and TOMM20 in CCD-1137Sk cells upon co-culturing with HT-29. With respect to the metabolic profile of HT-29 cells, 2-D cultures did not reveal any significant change, while in Dynarrays, the data showed an upregulation of HK-2, TIGAR, LDH, and TOMM20. This could indicate that the signaling interplay in the 3-D cultures was more extensive, leading to a clearer response of the HT-29 cells. Furthermore, the lactate export carrier MCT4 was upregulated in fibroblasts in 2-D upon co-cultivation and it showed a prevalence over MCT1 in the fibroblasts of the Dynarray cultures. The mitochondrial membrane potential, as measured by the TMRM fluorescence intensity, was mostly low in fibroblasts. In contrast, the TMRM signal was higher but variable from cell to cell in HT-29 cells. Finally, chip-based 3-D co-cultures of HT-29 and CCD-1137Sk cells grew more robustly than the corresponding mono-cultures, although it was not possible to unequivocally attribute this to one of the two cell types. Altogether, these data pointed to a metabolic cross-talk between cancer and fibroblast cells in co-cultures, which primarily affected the metabolic behavior of the fibroblasts, and in Dynarrays, HT-29 cells as well. This result is most consistent with a scenario in which HT-29 cells induce CCD-1137Sk cells to switch to a glycolytic and autophagic state, to lose mitochondrial activity, and to gain lactate export capabilities. Conversely, HT-29 cells apparently remained mixed in terms of mitochondrial activity and the expression of lactate importers and exporters, suggesting the presence of both catabolic and anabolic states. This was nicely corroborated by the bimodal expression patterns of HK-2, TIGAR, and LC3. Similar features, likely responsible for the differential susceptibility to drug treatment, were previously observed in mouse models and tumor tissue [51,52]. In particular, metabolic heterogeneity of malignant cells [53] and bad prognosis together with decreased overall survival upon enhanced MCT4 expression of tumors are fitting to the present study [54]. Finally, HT-29 in Dynarray co-cultures also gained in HK-2, TIGAR, LDH, and TOMM20, consistent with a combination of high glucose and lactate metabolism capability [26] that could jointly fuel the production of pyruvate and oxidative phosphorylation.

While the metabolic interaction patterns were observed in monolayer and chip-based co-cultures, no effect of co-culturing was evident on spheroid growth. This might be explained by three not mutually exclusive scenarios: First, fibroblasts remained rather isolated and encapsulated in spheroid co-cultures while they were more loosely interspersed in the chip-based co-cultures. Thus, one might speculate that the distribution of paracrine signals between the tumor and fibroblast cells was not comparable between the two culture conditions. Second, it might be that the relatively large spheroids of up to 1000 µm in diameter created more profound gradients of oxygen, nutrients, and waste products than the 300-µm-wide Dynarray-chip cultures. This might have intrinsically changed the basic metabolic profile of these cells or their propensity to react to each other. Finally, it cannot be completely excluded that the measurement of the spheroid diameter was insufficient to detect metabolic alterations induced upon co-cultivation. For this reason, the seeding densities in experiments focusing on the metabolic aspects of spheroid cultures had to be altered in a way that ensured a homogeneous size distribution as compared to the 300-µm Dynarray cavities. More analyses should be undertaken to reveal changes in the metabolic state of the cells. However, even if these existed, they were apparently so subtle that they had no robust effect on cell growth as observed in the chip format.

## 5. Conclusions

In summary, this work described different cellular co-culture models to mimic mutual metabolic interactions between HT-29 cancer and CCD-1137Sk fibroblast cells. In particular, the differential expression patterns of MCT1, MCT4, HK-2, TIGAR, TOMM20, LC3, and P62 in HT-29 and CCD-1137Sk cells, as well as the low mitochondrial membrane potential in fibroblasts as observed in mono-layer 2-D and chip-based 3-D co-cultures were consistent with reverse Warburg-type interactions. Together with the additional synergistic effects of co-culturing on cell proliferation in Dynarray chips, these features were comparable to those observed in tumor tissue. In contrast, spheroid co-cultures did not show such effects. In the future, the presented models might be employed for addressing basic molecular signaling and cell physiology aspects, as well as for testing metabolic modifiers, in combination with anticancer drugs for the treatment of neoplastic disorders.

## Figures and Tables

**Figure 1 cells-09-01900-f001:**
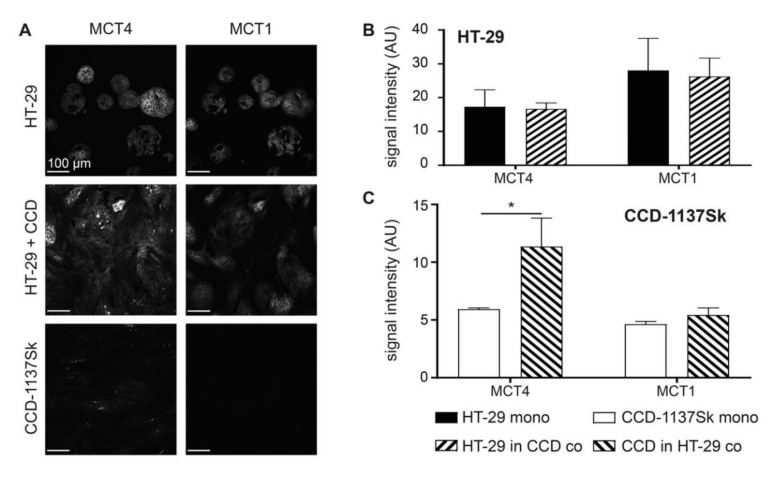
Monolayer co-cultures of HT-29 and CCD-1137Sk show enhanced expression of mono-carboxylate transporters, MCT4 in fibroblasts. HT-29 and CCD-1137Sk cells were either seeded alone or in co-culture and grown to a sub-confluent state for four days. Then, cells were fixed and stained with diamidino-2-phenylindol (DAPI) as well as antibodies against MCT4 and MCT1 as markers for nuclei, lactate export, and lactate import, respectively (**A**–**C**). (**A**) Representative confocal images of fluorescence staining for markers and cultures as indicated. (**B**) and (**C**) Graphs show quantitative analysis of the fluorescence intensity values for markers and cell type as indicated. Mean + SEM (*n* = 3 experiments; * *p* < 0.05).

**Figure 2 cells-09-01900-f002:**
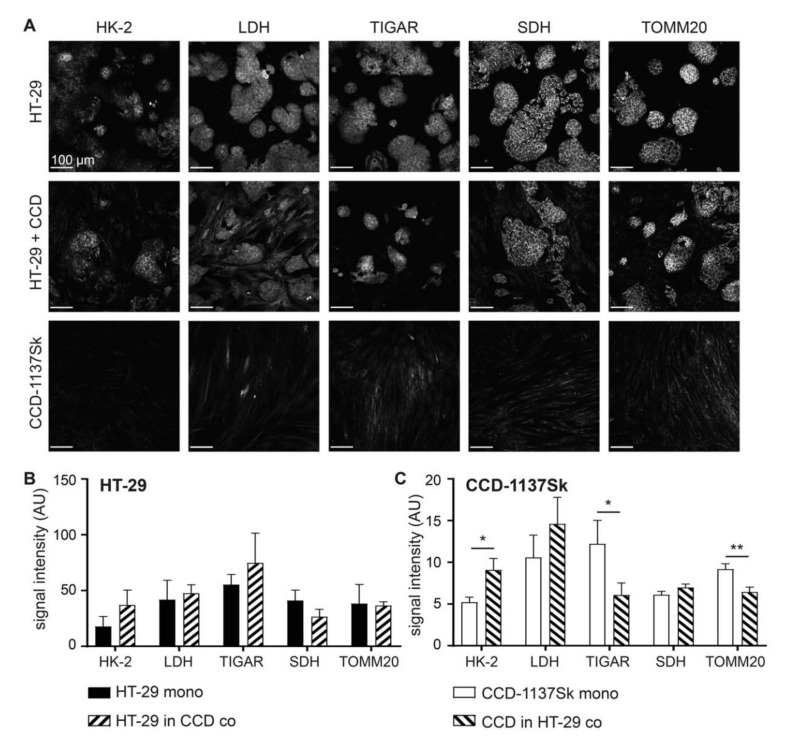
Fibroblasts in monolayer co-cultures of HT-29 and CCD-1137Sk show enhanced glycolytic and reduced oxidative phosphorylation markers. HT-29 and CCD-1137Sk cells were either seeded alone or in co-culture and grown to a sub-confluent state for four days. Then, cells were fixed and stained with DAPI as well as antibodies against HK-2, lactate dehydrogenase (LDH), TP53-induced glycolysis and apoptosis regulator (TIGAR), succinate dehydrogenase (SDH), and: translocase of outer mitochondrial membrane 20 (TOMM20) as markers for nuclei, glucose breakdown, pyruvate-lactate metabolism, negative glycolysis regulation, oxidative phosphorylation, and mitochondrial content, respectively (**A–C**). (**A**) Representative confocal images of fluorescence staining for markers and cultures as indicated. B and (**C**) Graphs show quantitative analysis of fluorescence intensity values for markers and cell types as indicated. Mean + SEM (*n* = 3 experiments; * *p* < 0.05, ** *p* < 0.01).

**Figure 3 cells-09-01900-f003:**
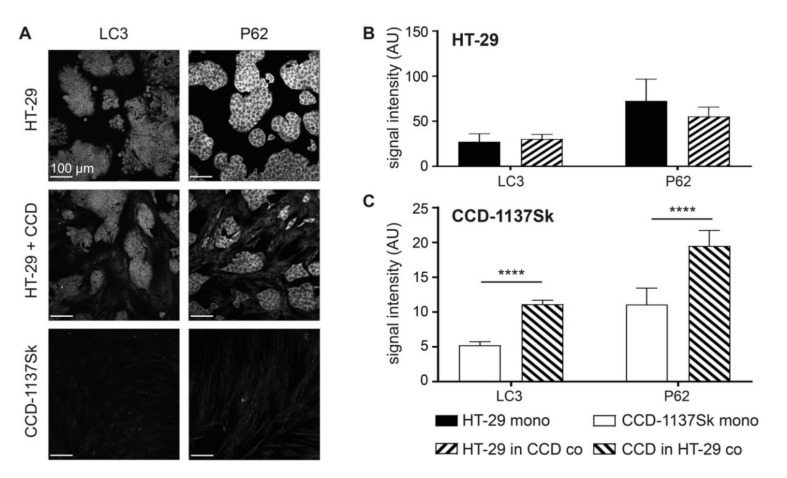
Fibroblasts in monolayer co-cultures of HT-29 and CCD-1137Sk display enhanced levels of autophagy markers. HT-29 and CCD-1137Sk cells were either seeded alone or in co-culture and grown to a sub-confluent state for four days. Then, cells were fixed and stained with DAPI as well as antibodies against LC3 and P62 as markers for nuclei and autophagy (**A–C**). (**A**) Representative confocal images of fluorescence staining for markers and cultures as indicated. (**B**) and (**C**) Graphs showing quantitative analysis of fluorescence intensity values for markers and cell type as indicated. Mean + SEM (*n* = 3 experiments; **** *p* < 0.0001).

**Figure 4 cells-09-01900-f004:**
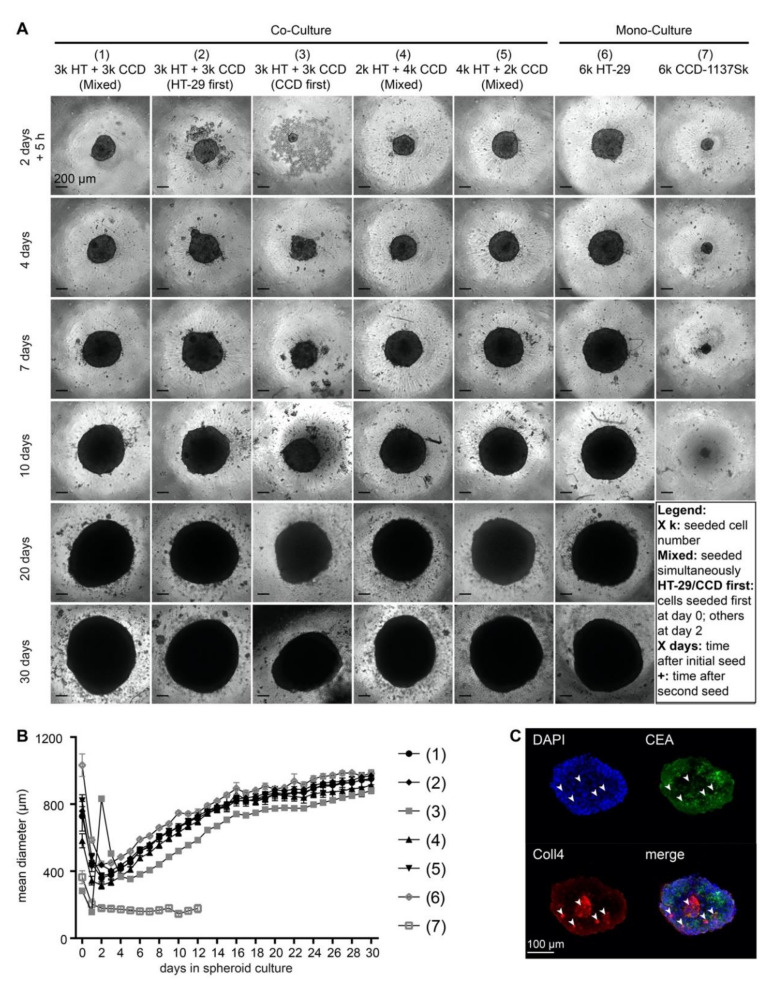
Growth of HT-29 spheroids is unaffected in the presence of CCD-1137Sk cells. Human HT-29 or CCD-1137Sk cells were seeded into ultra-low attachment plates to form spheroids, following different seeding orders (mixed, HT-29 first or CCD-1137Sk first) and cell numbers (ranging from 2000 to 6000 cells per type) as indicated. From 0 to 30 days in culture, brightfield images were taken to perform spheroid-size analysis based on the mean diameter (**A–B**). Additionally, co-culture spheroids with 250 HT-29 cells together with simultaneously seeded 500 CCD-1137Sk cells were fixed after four days and stained with DAPI as well as with anti-CEA and anti-Coll4 antibodies as markers for nuclei, HT-29 cells, and fibroblasts, respectively (**C**). (**A**) Representative brightfield images of HT-29 and CCD-1137Sk mono- and co-culture spheroids seeded in different ratios and orders, as depicted. (**B**) Graph shows a quantitative analysis of the mean diameters of mono- and co-culture spheroids at different time-points as indicated. Mean + SEM (*n* = 5–9 experiments). (**C**) Representative optical slice of DAPI (blue), anti-CEA (green), and anti-Coll4 signals (red). Arrowheads, exemplary clusters of Coll4-positive areas. Bottom right panel shows an overlay image of all three channels.

**Figure 5 cells-09-01900-f005:**
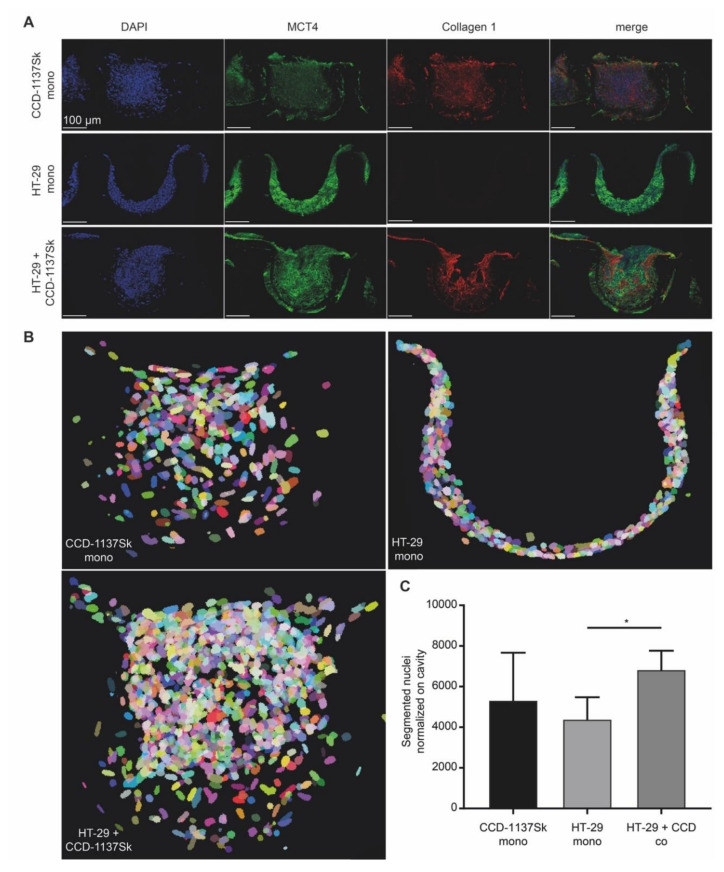
Chip-based 3-D cultures reveal morphological and metabolic changes upon co-cultivation of HT-29 and CCD-1137Sk cells. A total of 4000 cells per chip cavity were seeded to yield either mono- or co-cultures of HT-29 and CCD-1137Sk cells. In the case of co-cultures, 2000 HT-29 cells were co-seeded with 2000 CCD-1137Sk cells per well. After four days, chips were fixed, transversally sectioned, and then stained with DAPI as well as antibodies against MCT4 and collagen 1 as markers for nuclei, lactate export, and extracellular matrix (ECM), respectively. Subsequently, sections were imaged with confocal microscopy. (**A**) Representative maximum-z projections. (**B**) Representative display of segmented nuclei from different culture conditions as indicated. Pseudo-colors were chosen to discriminate between individual nuclei. They do not indicate different cell types. (**C**) Quantitative analysis of nuclear counts. Mean + SEM (*n* = 3 experiments).

**Figure 6 cells-09-01900-f006:**
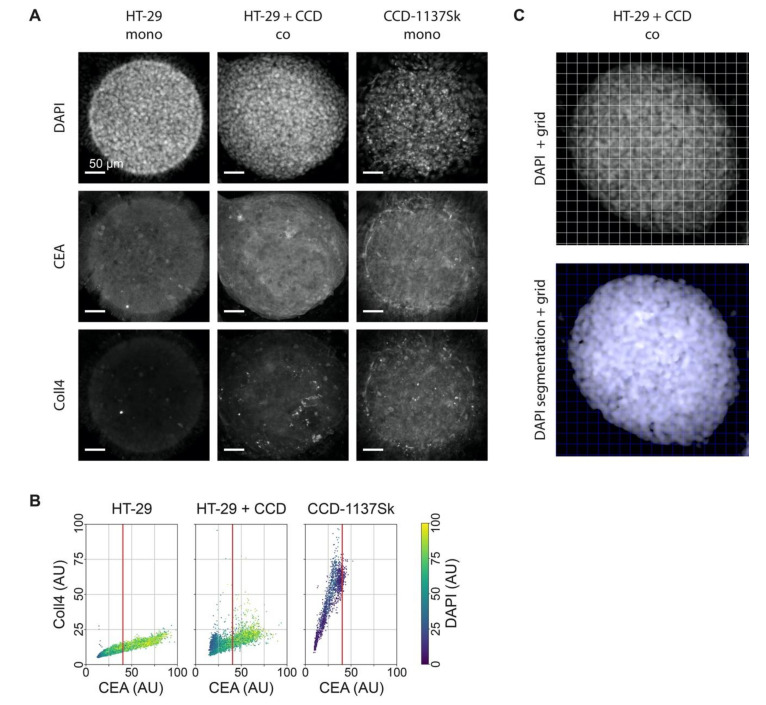
DAPI, CEA, and Coll4 staining jointly discriminate HT-29 and CCD-1137Sk cell populations in 3-D Dynarray co-cultures. A total of 4000 cells per chip cavity were seeded to yield either mono- or co-cultures of HT-29 and CCD-1137Sk cells. In case of co-cultures, 2000 HT-29 cells were co-seeded with 2000 CCD-1137Sk cells per well. After four days, chips were fixed, cleared, and stained with DAPI and antibodies against CEA and Coll4 as markers for nuclei, cancer cells, and fibroblasts, respectively. Subsequently, samples were imaged with 3-Dconfocal microscopy. (**A**) Representative sum-z projections. (**B**) Scatter plots showing the anti-Coll4 intensity distribution as a function of the anti-CEA intensity for HT-29 mono-cultures (left), HT-29 + CCD-1137Sk co-cultures (middle), and CCD-1137Sk mono-cultures (right). Each dot shows the average intensity of voxel cubes in a total of five Dynarray cavities. Red vertical line, CEA brightness range that contains 95% of all CCD-1137Sk voxel cubes in mono-culture. (**C**) Exemplary images showing the voxel-cube grid used for the quantification of the average fluorescence intensities. Upper panel, raw sum-z projection of DAPI signal; lower panel, sum-z projection upon nuclei segmentation and background subtraction.

**Figure 7 cells-09-01900-f007:**
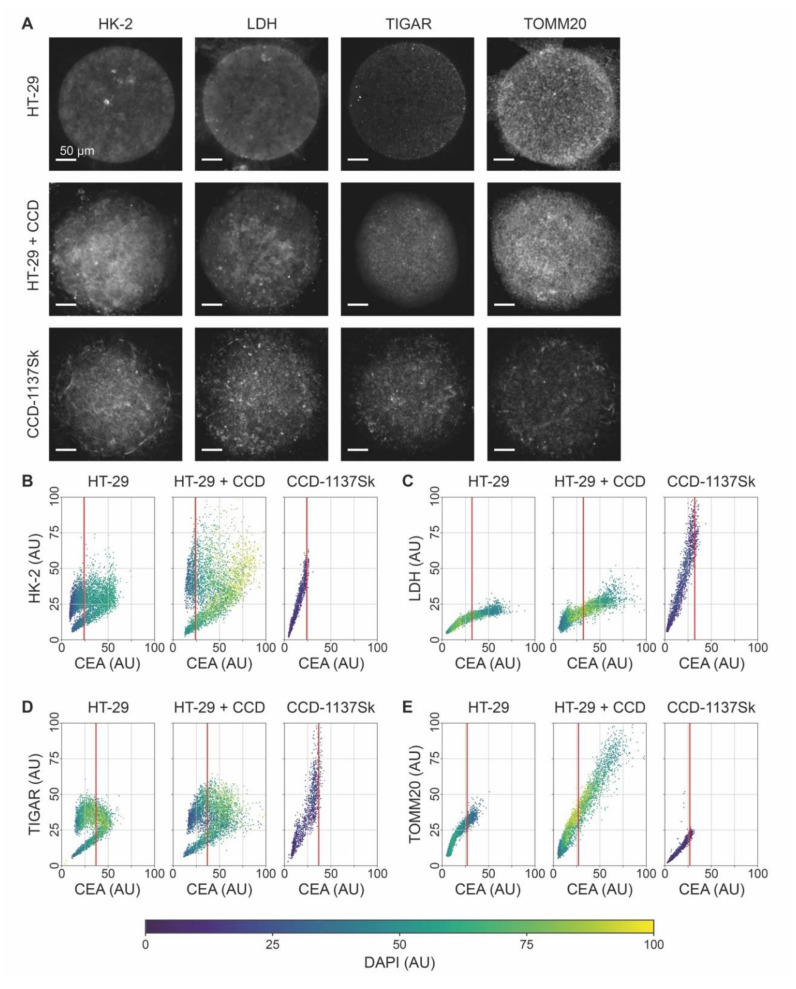
The expression of relevant metabolic marker proteins is altered upon co-culturing of HT-29 and CCD-1137Sk cells in Dynarrays. A total of 4000 cells per chip cavity were seeded to yield either mono- or co-cultures of HT-29 and CCD-1137Sk cells. In the case of co-cultures, 2000 HT-29 cells were co-seeded with 2000 CCD-1137Sk cells per well. After four days, chips were fixed, cleared, and stained with DAPI and antibodies against CEA as well as HK-2, LDH, TIGAR, or TOMM20 as markers for nuclei, cancer cells, and different metabolic activities, respectively. Then, samples were imaged with 3-D confocal microscopy. (**A**) Representative sum-z projections. (**B**–**E**) Scatter plots showing the fluorescence intensity distributions of anti-HK-2 (**B**), anti-LDH (**C**), anti-TIGAR (**D**), or anti-TOMM20 (**E**) as a function of the anti-CEA intensity for HT-29 mono-cultures (left), HT-29 + CCD-1137Sk co-cultures (middle), and CCD-1137Sk mono-cultures (right). Each dot shows the average intensity of a voxel cube in a total of five Dynarray cavities. Red vertical line, CEA brightness range that contains 95% of all CCD-1137Sk voxel cubes in mono-culture.

**Figure 8 cells-09-01900-f008:**
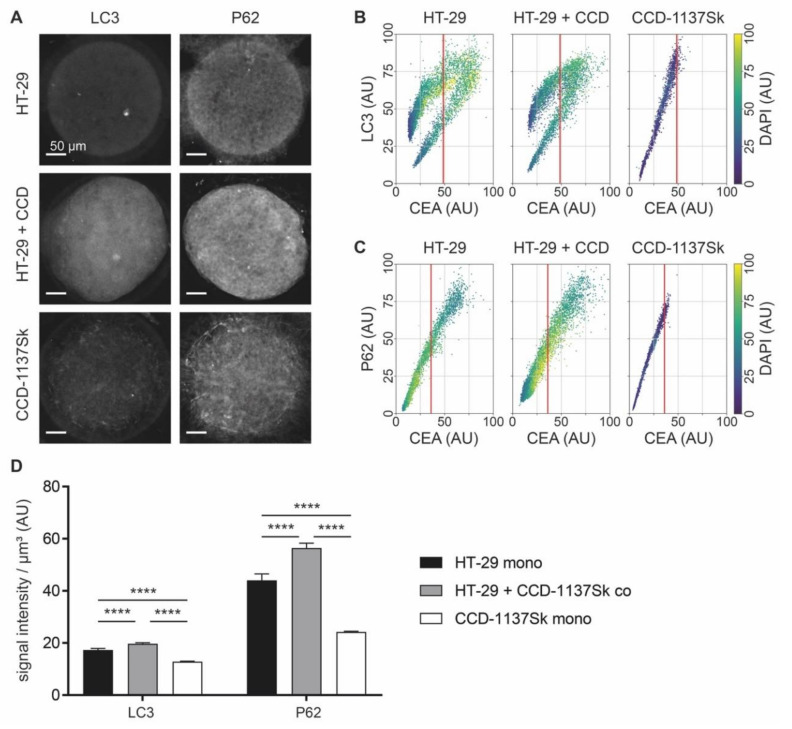
Co-culturing of HT-29 and CCD-1137Sk cells in Dynarrays leads to a rise of autophagic markers. A total of 4000 cells per chip cavity were seeded to yield either mono- or co-cultures of HT-29 and CCD-1137Sk cells. In the case of co-cultures, 2000 HT-29 cells were co-seeded with 2000 CCD-1137Sk cells per well. After four days, chips were fixed, cleared, and stained with DAPI and antibodies against CEA as well as LC3 or P62 as markers for nuclei, cancer cells, and autophagy, respectively. Then, samples were imaged with 3-D confocal microscopy. (**A**) Representative sum-z projections. (**B**,**C**) Scatter plots showing fluorescence intensity distributions of anti-LC3 (**B**) or anti-P62 signals (**C**) as a function of anti-CEA intensity for HT-29 mono-cultures (left), HT-29 + CCD-1137Sk co-cultures (middle), and CCD-1137Sk mono-cultures (right). Each dot shows the average intensity of a voxel cube of a total of five Dynarray cavities. Red vertical line, CEA brightness range that contains 95 % of all CCD-1137Sk voxel cubes in mono-culture. (**D**) LC3/P62 fluorescence intensity per µm^3^. Mean + SEM (*n* = 5 experiments).

**Figure 9 cells-09-01900-f009:**
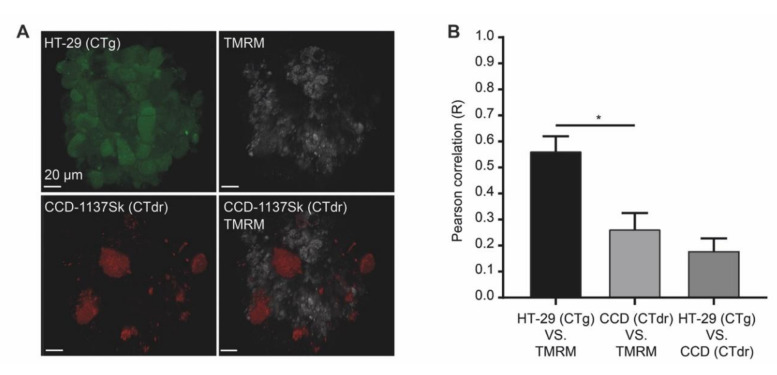
Chip-based 3-D cultures reveal a low mitochondrial membrane potential in fibroblasts upon co-cultivation of HT-29 and CCD-1137Sk cells. Human HT-29 and CCD-1137Sk cells were live stained with CTg and CTdr, respectively, and then co-seeded in Dynarray chips for four days. To evaluate the mitochondrial membrane potential, TMRM was analyzed in 3-D live cell imaging. (**A**) Representative maximum-z projections of fluorescence signals for CTg, CTdr, and TMRM, as indicated. (**B**) Quantitative analysis of Pearson’s correlation between CT-dyes and TMRM as a measure of cell-type-specific TMRM fluorescence intensity. Mean + SEM (*n* = 3 experiments).

**Figure 10 cells-09-01900-f010:**
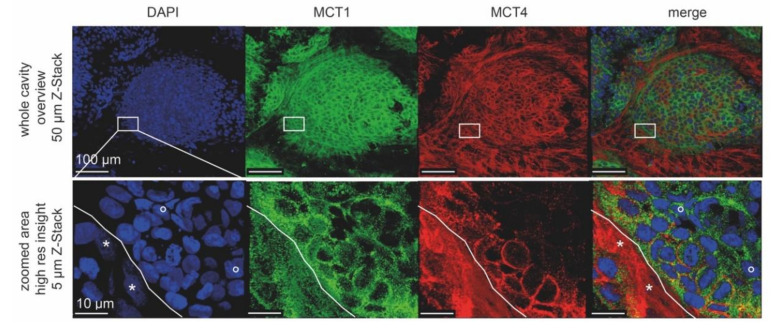
High-resolution confocal images reveal differential cell-type-specific expression of lactate shuttle proteins upon co-cultivation of HT-29 and CCD-1137Sk cells. Per chip cavity, 2000 HT-29 and 2000 CCD-1137Sk cells were co-seeded. After four days, cultures were fixed, cleared, and then stained with DAPI as well as antibodies against MCT4 and MCT1 as markers for nuclei, lactate export, and lactate import, respectively. Subsequently, samples were imaged with confocal microscopy. Shown are representative maximum-z projections with the indicated thickness. Lower panels were taken from the same specimen region as boxed in the upper panels. White solid lines in the lower panels indicate the border between a zone primarily populated by CCD-1137Sk (lower left) and HT-29 cells (upper right). Asterisks and open circles point to typically dim and large nuclei of CCD-1137Sk vs. bright and small nuclei of HT-29 cells.

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
