# Peer review of "A Scaffold-Free 3-D Co-Culture Mimics the Major Features of the Reverse Warburg Effect In Vitro"

_cells, 2020, doi:10.3390/cells9081900_

Round 1

Reviewer 1 Report

Authors investigated in chip based fibroblast co-cultures with HT-29 colon cancer cells in three-dimensional form, where 3D growth of HT-29 tumor cells was facilitated, and a dynamic cross-talk between fibroblasts and tumor cells affected cell growth and metabolism.

Authors investigated the reverse Warburg hypothesis, which included signaling from cancer cell to stromal fibroblast which modulates the metabolic behavior and induces a shift to aerobic glycolysis. Tumor cells and stroma fibroblasts construct a functional unit, whose metabolic activity should not be investigated separately.

The Introduction contains important backgroud data, describes the need of the study and declares the aims.

The methods are provided in sufficient details, including the statistic analysis.

The Results are provided in appropriate details, and their presentation is also good quality.

The Discussion concentrates on the own results and discuss them in relation to the relevant literature background.

Author Response

We are grateful for the overall positive response.

Reviewer 2 Report

The work is convincing the data are clearly, in my opinion it can be published as it is in the current version.

Author Response

We are grateful for the rather positive response.

Reviewer 3 Report

This study aims to compare metabolic characteristics of co-cultures set up in different in vitro systems and concludes that a Chip-based co-culture setup was able to model the reverse Warburg effect.

Major points that would include the quality of the manuscript:

The Introduction should set out the different culture systems used and explain their similarities and differences, especially comparing the standard 3D spheroid and the chip-based method. 

In Figure 2A, there seems to be a large change in spheroid volume between day 20 and day 30, yet this does not seem to be reflected in the graphs in Figure 2B.

Figure 2C, the structures that the arrow-heads are meant to be pointing at are not clear.  Perhaps higher magnification images are needed.  However, in addition, it is not clear how the authors know which cell-type these structures represent, as at this point in the manuscript, the individual populations of cells are not distinctly labelled.

The chip-based culture system seems to involve seeding in collagen and yet no comparison with e.g. 2D seeding onto collagen is provided.

In Figure 3B, it is not clear how the pseudo-colouring was created and on what basis.  i.e. what do the different colours represent? Again, if they are meant to represent different populations of cells, how were they identified?

Figure 4C: the authors acknowledge that the cell tracker dyes have an effect on the expression of the metabolic markers.  Yet, they go on to use them in later experiments which makes it hard to interpret what is the effect of the use of the dyes and what is the effect of the co-culture.

Figure 5 - hard to see the TMRM in the merge - perhaps would be clearer to show the merge between TMRM and just the fibroblast marker?

Minor points:

There are some grammatical errors that should be corrected

There are also some unusual terms used e.g. 'anticlimax' to indicate growth saturation.

The abbreviation DIV is not helpful.  It is used to indicate 'days in vitro' but surely the cells were in vitro before seeding for individual experiments.  The usual indication of timepoints would be clearer.

Author Response

This study aims to compare metabolic characteristics of co-cultures set up in different in vitro systems and concludes that a Chip-based co-culture setup was able to model the reverse Warburg effect.

Major points that would include the quality of the manuscript:

The Introduction should set out the different culture systems used and explain their similarities and differences, especially comparing the standard 3D spheroid and the chip-based method. 

>> We agree. New chapters on the comparison of different culture systems have been added in the introduction on lines 78-89 and in the discussion on lines 504-537.

In Figure 2A, there seems to be a large change in spheroid volume between day 20 and day 30, yet this does not seem to be reflected in the graphs in Figure 2B.

>> Apparently, something went wrong here in the end-formatting of this figure. This has now been corrected (new Figure 4). We are sorry for the error.

Figure 2C, the structures that the arrow-heads are meant to be pointing at are not clear.  Perhaps higher magnification images are needed.  However, in addition, it is not clear how the authors know which cell-type these structures represent, as at this point in the manuscript, the individual populations of cells are not distinctly labelled.

>> Apparently, something went wrong here in the end-formatting of this figure. This has now been corrected (new Figure 4). As for the different cell populations, these have now been differentiated by a set of three criteria: a) molecular markers (CEA, Coll4), b) localization in dense clusters of HT-29 cells, c) morphological features of nuclei. These features have been explained in the revised manuscript at lines 211-219 and depicted in new Figure S1 (2D) and Figure 6 (chip).

The chip-based culture system seems to involve seeding in collagen and yet no comparison with e.g. 2D seeding onto collagen is provided.

>> A comparison between 2D seeding in the absence and presence of collagen and its effects on our panel of tested markers has now been included in the new Figure S6 and in the discussion at lines 546-548. Apparently, collagen coating did not affect any of the assessed parameters in a significant manner.

In Figure 3B, it is not clear how the pseudo-colouring was created and on what basis.  i.e. what do the different colours represent? Again, if they are meant to represent different populations of cells, how were they identified?

>> Pseudo-colors were simply chosen to easily discriminate between the individual segmented nuclei. They were not meant to identify different cell populations. A corresponding remark has been placed in the corresponding figure legend (new Figure 5).

Figure 4C: the authors acknowledge that the cell tracker dyes have an effect on the expression of the metabolic markers.  Yet, they go on to use them in later experiments which makes it hard to interpret what is the effect of the use of the dyes and what is the effect of the co-culture.

>> We agree to this concern. Therefore, except for the TMRM measurement, we have omitted the data based on cell tracker dyes and have, instead, used a cell-type discrimination procedure based on CEA and Coll4 for the marker staining experiments. The new, extended data corresponding to the old Fig. 4 can now be found in Figures 6-8.

Figure 5 - hard to see the TMRM in the merge - perhaps would be clearer to show the merge between TMRM and just the fibroblast marker?

>> This was corrected as suggested (new Figure 9A).

Minor points:

There are some grammatical errors that should be corrected.

>> Corrected.

There are also some unusual terms used e.g. 'anticlimax' to indicate growth saturation.

>> Corrected.

The abbreviation DIV is not helpful.  It is used to indicate 'days in vitro' but surely the cells were in vitro before seeding for individual experiments.  The usual indication of timepoints would be clearer.

>> Corrected.

Reviewer 4 Report

This work probes in vitro a relevant effect existing in tumors. The hypothesis of the study is valid and the authors showed data that, in part, confirmed their hypothesis. The major drawback of this study is the comparison of these 2D and 3D gel-free cultures with matrigel cultures as controls. Although matrigel is not a suitable material, as also suggested in the introduction, it is the state of the art go-to gel used in cancer research. Therefore, comparing the obtained results in chip cultures against cells grown in Matrigel would have provided insightful information to the readers. Besides, the discussion is missing to compare the obtained results with many other excellent works on in vitro cancer cell culture using 3D matrices other than Matrigel. Most importantly, the authors need to explain in more detail why gel-free cultures are advantageous compared to gel-based cultures. Currently, in the materials science and tissue engineering communities, the most successful studies in cancer research use 3D cell cultures in natural and/or synthetic matrices.

Author Response

We agree to these concerns. Accordingly, BME-based 3D cultures were made and several of the tested parameters were assessed. These new data can be found in the new Figure S5. Also, the discussion was substantially modified to fit the requested aspects in, please refer to lines 504-555.

Reviewer 5 Report

General Comments:

In this study, the authors set out to examine the metabolic interplay between cancer cells and fibroblasts using 2D, spheroid, and Dynarray chip assays. They used human HT-29 colon cancer cells and CCD-1137Sk fibroblasts as examples in co-cultures or separate cultures to address how co-culture of the two types of cells affected the metabolic status of the cancer cells or fibroblasts by means of immunofluorescence (IF) staining of cellular protein markers.

Although the study was aimed “to establish a BME/Matrigel-free 3D microarray cancer model to recapitulate the metabolic interplay between cancer and stromal cells that allows mechanistic analyses and drug testing in parallel,” it only used the established 2D or 3D culture systems. Not as the title stated, the major features of the reverse Warburg effect were not sufficiently examined in the experiments. The metabolic interactions between the cancer cells and the fibroblasts was not adequately addressed. Overall, there is a need to use additional cell lines/types to verify the findings and support the conclusions. There is no direct measurement of the metabolic changes of either the cancer cells or the fibroblasts in the co-culture system. Certain metabolism-related assays should be considered to be used to address the key claims. The single autophagy marker expression (via IF staining) does not support the biological action of autophagy, and MCT1 or MCT4 expression not necessarily reflects the import or export of lactate into or from the cells.

Specifics:

  1. In Fig 1A, the DAPI staining of CCD-1137Sk fibroblasts is almost invisible compared to that of the HT-29 cells. Were the images taken under the same imaging parameters? Even the expression of MCT1 and MCT4 were low in the CCD-1137Sk cells, the DAPI staining should be normally seen.

  1. In Fig 1B quantification of Fig 1A, it appears that the MCT-4 levels of the co-culture samples in the HT-29 group were lower than the mono-cultures, inconsistent with what the text described. How were the fluorescent signals of the CCD-1137Sk fibroblasts and the HT-29 cells separately quantified without the influence from the other type of cells in the co-cultures?

  1. The LC3 expression in the co-culture shown in Fig 1C was higher than the HT-29 mono-culture, and yet its quantification in Fig 1D was lower. In addition, could the upregulation of LC3 in the co-culture be an additive effect from both the CCD-1137Sk and the HT-29 cells?

  1. Fig 1E: How the LC3 SDS-PAGE images were detected and quantified? Does the LC3 I or LC3 II band represent a combined level of the protein from both the CCD-1137Sk and the HT-29 cells or only from the CCD-1137Sk cells? Were the cells sorted out and then Western blotting conducted after protein extraction? It seems that the LC3 I level was also much higher in the co-culture samples but did not mention or discuss. Additionally, the LC3 I band for the co-culture lane appeared to be at higher molecular weight compared to the CCD and HT-29 lanes. Was LC3 I modified in the co-culture conditions?

  1. The asterisks in Fig 1B, 1D, and 1F for comparisons were not defined in either the figure legends or the methods.

  1. Why Fig 2A #7 CCD-1137Sk cell mono-culture had no 20 DIV and 30 DIV cultures? Why the #3 co-cultures were much smaller till the 20 DIV but then became similar to the others on 30 DIV? This is not consistent with the statement for Fig 2 that the growth of the HT-29 cell spheroids was not affected by the presence of the CCD-1137Sk cells, especially when compared the #3 with the #6 samples. The sizes of the #5 samples were substantially larger than those of the #3 or #4 samples.

  1. Fig 2C: How could the fibroblast islets indicated by the arrows be defined and differentiated from the cancer cells in the co-cultures? The CellTracker method used in the Fig 4 and Fig 5 could be helpful in this experiment.

  1. In the Dynarray Chip quantification data shown in Fig 3C, was the nuclei count for the co-culture an additive value of the HT-29 and the CCD-1137Sk cells? If so, the co-culture nuclei count total was actually lower than that of adding the HT-29 and the CCD-1137Sk mono-cultures.

  1. Line 239: For checking co-culture impact on metabolic alterations, why using the autophagy marker LC3 and IF staining instead of measuring metabolic profiles of the cells?

  1. Fig 4A: The authors noticed that the CellTrackers had an impact on autophagic activity. This experimental caveat was not further discussed, and the technique was further used in the experiments corresponding to Fig 5, where mitochondrial membrane potential was examined. How the results in Fig 5 could be justified if the “labelling itself” affects autophagic phenotypes?

  1. In Fig 4C, it seems the CTr and LC3 were all red channel images, were their wavelengths overlapping and interfering with each other? Fig 4C and Fig S1 IF color coding were not consistent, and LC3 specific staining for the CCD-1137Sk cells wasn’t convincing. Is this section claiming for metabolic changes as stated in the beginning of the 3.4. text or addressing autophagy phenotype? Either way, additional experiments should be done to verify the results.

  1. Fig 5A: It seems the HT-29 cells outnumbered the CCD-1137Sk fibroblasts. If the two cell types were seeded at an equal initial number, did the phenotype mean that the cancer cells gain growth advantage over the fibroblasts within the culturing period?

  1. Line 273-274: It was described that “fibroblasts were mostly lacking measurable mitochondrial membrane potential.” Why was it the case? Was it experimental setup-specific or was it a common phenotype even if tested with additional cell lines?

  1. Line 274-275: It was stated that “regions outside of the circular entry of the chip cavities, which were primarily populated by fibroblasts.” Could the authors explain why the fibroblasts preferred to line up outside the circular entry although they were mixed with the HT-29 cells initially in the co-culture? This phenotype is also inconsistent with that shown in Fig 3, where the HT-29 cell mono-culture samples were distributed in the surrounding areas of the cavities of the chips. Moreover, the MCT4 expression in both the HT-29 and CCD-1137Sk cells was comparable in Fig 3.

  1. The Fig 6A was not described in the result section, and the differences between the 2D and 3D cultures were not compared or described in the results.

  1. Fig 6: How could the different cell populations (HT-29 and CCD-1137Sk) be differentiated? How the MCT1 and MCT4 staining for the HT-29 and CCD-1137Sk cell populations were discerned? Why the HT-29 and CCD-1137Sk cells stained for both MCT1 and MCT4 in the circular regions but not in the surrounding areas?

  1. Line 311-312, the cited work, “the expression of MCT4 and the lactate import carrier MCT1 are enhanced in malignant cells [34], regardless of the absence or presence of fibroblasts,” does not necessarily support the observation shown in Fig 6B although the staining of the center part of the samples was positive for both MCT4 and MCT1 since the staining could hardly be differentiated based on the cell types.

  1. Line 314-315: It was described that “chip-based 3D co-cultures of HT-29 and CCD-1137Sk cells grew more robustly than corresponding mono-cultures.” However, this statement was not supported by direct data, where cell growth or proliferation was measured. Although the nuclear counts were displayed in Fig 3C, how the counting was conducted and what type (or both types) of the cells grew better in the co-cultures were not clear.

  1. Line 316-319: “these data pointed to a metabolic cross-talk between cancer and fibroblast cells in co-cultures, which primarily affected the metabolic behavior of the fibroblasts. Indeed, this result is most consistent with a scenario in which HT-29 cells induced CCD-1137Sk cells to switch to an autophagic state, to lose mitochondrial activity, and to gain lactate export capabilities.” These claims were not based on metabolic or functional data but only a single marker immunofluorescence staining, and therefore should be supported by additional experiments.

  1. Typo: Line 323, “[37”

  1. Since the “low mitochondrial membrane potential in fibroblasts” was observed in both the “2D and chip-based 3D co-cultures” (line 342-344), why would it be necessary to use the 3D co-cultures to address the reverse Warburg effect, which was actually not explored in the experiments.

  1. Grammar and punctuation in the text should be thoroughly checked.

Author Response

General Comments:

In this study, the authors set out to examine the metabolic interplay between cancer cells and fibroblasts using 2D, spheroid, and Dynarray chip assays. They used human HT-29 colon cancer cells and CCD-1137Sk fibroblasts as examples in co-cultures or separate cultures to address how co-culture of the two types of cells affected the metabolic status of the cancer cells or fibroblasts by means of immunofluorescence (IF) staining of cellular protein markers.

Although the study was aimed “to establish a BME/Matrigel-free 3D microarray cancer model to recapitulate the metabolic interplay between cancer and stromal cells that allows mechanistic analyses and drug testing in parallel,” it only used the established 2D or 3D culture systems. Not as the title stated, the major features of the reverse Warburg effect were not sufficiently examined in the experiments. The metabolic interactions between the cancer cells and the fibroblasts was not adequately addressed. Overall, there is a need to use additional cell lines/types to verify the findings and support the conclusions. There is no direct measurement of the metabolic changes of either the cancer cells or the fibroblasts in the co-culture system. Certain metabolism-related assays should be considered to be used to address the key claims. The single autophagy marker expression (via IF staining) does not support the biological action of autophagy, and MCT1 or MCT4 expression not necessarily reflects the import or export of lactate into or from the cells.

Specifics:

  1. In Fig 1A, the DAPI staining of CCD-1137Sk fibroblasts is almost invisible compared to that of the HT-29 cells. Were the images taken under the same imaging parameters? Even the expression of MCT1 and MCT4 were low in the CCD-1137Sk cells, the DAPI staining should be normally seen.

>> As a matter of fact, all images were taken using the same parameters and, thus, DAPI staining of CCD-1137Sk cells showed dimmer and larger nuclei as compared to HT-29 cells, both in mono- as well as in co-culture. This feature was partially used to discriminate between CCD-1137Sk and HT-29 cells, although we have now included also further markers (see e.g. response to comment 2, 7, 10, 11).

  1. In Fig 1B quantification of Fig 1A, it appears that the MCT-4 levels of the co-culture samples in the HT-29 group were lower than the mono-cultures, inconsistent with what the text described. How were the fluorescent signals of the CCD-1137Sk fibroblasts and the HT-29 cells separately quantified without the influence from the other type of cells in the co-cultures?

>> Regarding the MCT-4 levels, more representative pictures have been included, now. As for the different cell populations, these have now been differentiated by a set of three criteria: a) molecular markers (CEA, Coll4), b) localization in dense clusters of HT-29 cells, c) morphological features of nuclei. These features have been explained in the revised manuscript at lines 211-219 and depicted in Fig. S1.

  1. The LC3 expression in the co-culture shown in Fig 1C was higher than the HT-29 mono-culture, and yet its quantification in Fig 1D was lower. In addition, could the upregulation of LC3 in the co-culture be an additive effect from both the CCD-1137Sk and the HT-29 cells?

>> We agree, that the LC3 immunofluorescence signal intensity in HT-29 cell clusters of the old Fig. 1C was apparently higher than in HT-29 monocultures, being in conflict with the quantitative analysis. In the new corresponding Figure 3, this error was corrected and more representative image pairs were chosen. As the quantification was redone with new samples and this resulted very similar to the old data, we think it should be correct.

  1. Fig 1E: How the LC3 SDS-PAGE images were detected and quantified? Does the LC3 I or LC3 II band represent a combined level of the protein from both the CCD-1137Sk and the HT-29 cells or only from the CCD-1137Sk cells? Were the cells sorted out and then Western blotting conducted after protein extraction? It seems that the LC3 I level was also much higher in the co-culture samples but did not mention or discuss. Additionally, the LC3 I band for the co-culture lane appeared to be at higher molecular weight compared to the CCD and HT-29 lanes. Was LC3 I modified in the co-culture conditions?

>> Clearly, the Western blot data as presented were not adequate for addressing the pertinent question in an appropriate manner. Since cell-type specific analysis is relevant for this manuscript and regular access to FACS was not possible during the revision period, we opted to leave Western blot data out altogether.

  1. The asterisks in Fig 1B, 1D, and 1F for comparisons were not defined in either the figure legends or the methods.

>> Fig 1 has been replaced by new figures 1-3. Asterisks have been defined in each legend now.

  1. Why Fig 2A #7 CCD-1137Sk cell mono-culture had no 20 DIV and 30 DIV cultures? Why the #3 co-cultures were much smaller till the 20 DIV but then became similar to the others on 30 DIV? This is not consistent with the statement for Fig 2 that the growth of the HT-29 cell spheroids was not affected by the presence of the CCD-1137Sk cells, especially when compared the #3 with the #6 samples. The sizes of the #5 samples were substantially larger than those of the #3 or #4 samples.

>> Apparently, something went wrong here in the end-formatting of this figure. This has now been corrected (new Fig. 4). We are sorry for the error.

  1. Fig 2C: How could the fibroblast islets indicated by the arrows be defined and differentiated from the cancer cells in the co-cultures? The CellTracker method used in the Fig 4 and Fig 5 could be helpful in this experiment.

>> To discriminate between HT-29 and CCD-1137Sk cells, we have now included immunostaining of CEA and Coll4. Please see new Figure 4C.

  1. In the Dynarray Chip quantification data shown in Fig 3C, was the nuclei count for the co-culture an additive value of the HT-29 and the CCD-1137Sk cells? If so, the co-culture nuclei count total was actually lower than that of adding the HT-29 and the CCD-1137Sk mono-cultures.

>> The quantification here was indeed an additive value. However, since the total number of cells seeded per well (4,000) was equal for each condition (i.e. 2,000 CCD + 2,000 HT-29 in co-cultures), the increase in nuclei should indeed be attributable to an enhanced proliferation in the co-cultures.

  1. Line 239: For checking co-culture impact on metabolic alterations, why using the autophagy marker LC3 and IF staining instead of measuring metabolic profiles of the cells?

>> We perfectly agree and we are thankful to this referee for raising this concern. It has allowed us to include now a new panel of additional markers (HK-2, LDH, SDH, TIGAR, TOMM20) that give a more detailed insight into the observed phenomena. These markers were also applied to the 2D cultures. We think, that this has helped to significantly improve this manuscript.

  1. Fig 4A: The authors noticed that the CellTrackers had an impact on autophagic activity. This experimental caveat was not further discussed, and the technique was further used in the experiments corresponding to Fig 5, where mitochondrial membrane potential was examined. How the results in Fig 5 could be justified if the “labelling itself” affects autophagic phenotypes?

>> This is a very good point. Indeed, we have omitted the use of CellTrackers now for most of the analyses. To identify cell-type specific effects, we have switched to a set of three criteria (location, molecular markers, nuclear morphology) that were, however, utilizable only in fixed samples. Thus, for the live-cell imaging, we had to rely on the CellTracker-based analysis. To make the setting transparent for the reader, we included a remark on the potential pitfalls of CellTracker labeling on lines 467-471.

  1. In Fig 4C, it seems the CTr and LC3 were all red channel images, were their wavelengths overlapping and interfering with each other? Fig 4C and Fig S1 IF color coding were not consistent, and LC3 specific staining for the CCD-1137Sk cells wasn’t convincing. Is this section claiming for metabolic changes as stated in the beginning of the 3.4. text or addressing autophagy phenotype? Either way, additional experiments should be done to verify the results.

>> As rightly suggested, these data needed improvement. Given the uncertainties, we opted to take out all CellTracker-based data except the TMRM experiment (see response to comment 10), because for live imaging we had to. As already outlined in our responses to concerns 2 and 7, we have now used a combined set of three criteria for cell-type identification: a) molecular markers (CEA, Coll4), b) localization in dense clusters of HT-29 cells, c) morphological features of nuclei. Also, we have significantly expanded the set of tested markers. Thus, the old Fig. 4 was replaced by new Figures 6-8.

  1. Fig 5A: It seems the HT-29 cells outnumbered the CCD-1137Sk fibroblasts. If the two cell types were seeded at an equal initial number, did the phenotype mean that the cancer cells gain growth advantage over the fibroblasts within the culturing period?

>> This is exactly what we think as well.

  1. Line 273-274: It was described that “fibroblasts were mostly lacking measurable mitochondrial membrane potential.” Why was it the case? Was it experimental setup-specific or was it a common phenotype even if tested with additional cell lines?

>> Regarding the first point, we can only speculate that mitochondria were either largely lacking in fibroblasts, not charged due reduced metabolic activity, or a combination of both. As for the second point, we are thankful for this advice and we have done a lot of work to complement this manuscript with data sets using co-culturing between breast cancer cells, MDA-MB-231, and CCD-1137Sk cells. These novel data can be found in Supplementary Figures 2-4. In essence, they nicely confirm the data obtained with HT-29 co-cultures. For the sake of time, the TMRM experiment was out of reach, but the TOMM20 data shown in Figure S3 suggest, that mitochondria content was reduced in CCD1137-Sk cells also in MDA-MB-231 co-cultures.

  1. Line 274-275: It was stated that “regions outside of the circular entry of the chip cavities, which were primarily populated by fibroblasts.” Could the authors explain why the fibroblasts preferred to line up outside the circular entry although they were mixed with the HT-29 cells initially in the co-culture? This phenotype is also inconsistent with that shown in Fig 3, where the HT-29 cell mono-culture samples were distributed in the surrounding areas of the cavities of the chips. Moreover, the MCT4 expression in both the HT-29 and CCD-1137Sk cells was comparable in Fig 3.

>> Certainly, these are interesting unexplained features of the chip cultures. When addressing the issue of differential location of cells in chips upon mono-/co-culture, it first needs to be mentioned that in co-cultures, fibroblasts were making the first-line contact with the chip surface all over the place (described in a novel part of the discussion on lines 548-553) and HT-29 were located more towards the inside and did rarely contact the chip wall itself. Conversely, in monocultures, HT-29 cells directly grew on the chip walls and made flat layers on top of each other. Thus, one could speculate that HT-29 were, in principle, able to grow directly on the chip walls, but if possible, they preferred to grow on a layer of fibroblasts. From the current perspective, this preference might be due to either physical, chemical, or biological factors. For example, it could be that the chip walls had a stiffness that HT-29 cells preferred to avoid (physical constraint). Alternatively, Collagen 1 might have been more attractive for CCD-1137Sk cells than for HT-29 (chemical constraint). Finally, the fibroblasts might have sent out some sort of signaling to make the HT-29 cells dwell more towards the center of the cavity (biological constraint). If requested, a similar kind of explanation could be included into the discussion. For the time being, we preferred to leave it out, because it might be too speculative at this point. With respect to the similar expression of MCT4, we agree and have now added a corresponding remark on lines 316f.

  1. The Fig 6A was not described in the result section, and the differences between the 2D and 3D cultures were not compared or described in the results.

>> Correct. Fig. 6A was taken out of the revised manuscript due to lack of relevance.

  1. Fig 6: How could the different cell populations (HT-29 and CCD-1137Sk) be differentiated? How the MCT1 and MCT4 staining for the HT-29 and CCD-1137Sk cell populations were discerned? Why the HT-29 and CCD-1137Sk cells stained for both MCT1 and MCT4 in the circular regions but not in the surrounding areas?

>> As already outlined in our responses to several comments, we have now used a combined set of three criteria for cell-type identification: a) molecular markers (CEA, Coll4), b) localization in dense clusters of HT-29 cells, c) morphological features of nuclei. With respect to the differential distribution of MTC1 and MCT4, we speculate that both, cell type and location might be important. At least in 2D-co-cultures, MCT1 was more expressed on HT-29 than MCT4, while for CCD-1137Sk cells it was the other way-round. This is also reflected in the border region between circular cavity and surrounding areas as shown in Fig. 10 lower panels and likely in the circular cavity itself, where patches of prevalent MCT1 or MCT4 signals were visible (Fig. 10 upper panels). However, in the surrounding area, i.e. far away from the circular area, MCT1 expression rapidly decreased. One could speculate that this was due to a more complete lack of HT-29 cells in this region or that fibroblasts in that zone were modulated to further downregulate MCT1.

  1. Line 311-312, the cited work, “the expression of MCT4 and the lactate import carrier MCT1 are enhanced in malignant cells [34], regardless of the absence or presence of fibroblasts,” does not necessarily support the observation shown in Fig 6B although the staining of the center part of the samples was positive for both MCT4 and MCT1 since the staining could hardly be differentiated based on the cell types.

>> This statement was taken out.

  1. Line 314-315: It was described that “chip-based 3D co-cultures of HT-29 and CCD-1137Sk cells grew more robustly than corresponding mono-cultures.” However, this statement was not supported by direct data, where cell growth or proliferation was measured. Although the nuclear counts were displayed in Fig 3C, how the counting was conducted and what type (or both types) of the cells grew better in the co-cultures were not clear.

>> We agree that the old version of the manuscript did not sufficiently describe the culture conditions and display modes. We hope, this has now been amended to better support the statement of a robust growth of co-cultures in chips. We think, the cited statement is correct, although we can certainly not attribute it to a specific cell type. To make this latter point, a short note has been added on lines 567f.

  1. Line 316-319: “these data pointed to a metabolic cross-talk between cancer and fibroblast cells in co-cultures, which primarily affected the metabolic behavior of the fibroblasts. Indeed, this result is most consistent with a scenario in which HT-29 cells induced CCD-1137Sk cells to switch to an autophagic state, to lose mitochondrial activity, and to gain lactate export capabilities.” These claims were not based on metabolic or functional data but only a single marker immunofluorescence staining, and therefore should be supported by additional experiments.

>> We are grateful to this referee for raising this concern. It has allowed us to include now a panel of additional markers (HK-2, LDH, SDH, TIGAR, TOMM20) that give a more detailed insight into the observed phenomena. These markers were also applied to the 2D cultures. We think, that this has helped to significantly improve this manuscript.

  1. Typo: Line 323, “[37”

>> Corrected.

  1. Since the “low mitochondrial membrane potential in fibroblasts” was observed in both the “2D and chip-based 3D co-cultures” (line 342-344), why would it be necessary to use the 3D co-cultures to address the reverse Warburg effect, which was actually not explored in the experiments.

>> This statement was taken out. As for the reason to use both, 2D and 3D cultures, one major reason, i.e. the lack of gradients of oxygen, waste products etc. was actually already mentioned. Next, in the light of the gradients just mentioned, even a lack of differences between 2D and 3D would be interesting to learn about relevant modulators. Finally, the novel analyses on different metabolic markers that were requested by this referee showed that cancer cells actually behaved differently in 3D as compared to 2D, while fibroblasts did not. Thus, this is a further argument for the double-check.

  1. Grammar and punctuation in the text should be thoroughly checked.

>> Done as requested.

Round 2

Reviewer 3 Report

The authors have addressed the questions and comments raised in the original review.

Reviewer 5 Report

N/A